# Effect of Different Climatic Regions and Seasonal Variation on the Antibacterial and Antifungal Activity, and Chemical Profile of *Helichrysum aureonitens* Sch. Bip

**DOI:** 10.3390/metabo12080758

**Published:** 2022-08-17

**Authors:** Wilson Bamise Adeosun, Olusola Bodede, Gerhard Prinsloo

**Affiliations:** 1Department of Agriculture and Animal Health, University of South Africa, Johannesburg 1709, South Africa; 2Department of Chemistry, University of Pretoria, Pretoria 0028, South Africa

**Keywords:** *Helichrysum aureonitens*, environmental effects, seasonal variation, antifungal, antibacterial, metabolomics, ^1^H-NMR, medicinal plants

## Abstract

Native South Africans make use of *Helichrysum aureonitens* Sch. Bip. extracts for the treatment of a variety of infections and they are important in traditional medicinal preparations. This study investigated the effect of seasonal variation and geographical location on the antibacterial and antifungal activities of *H. aureonitens.* Material was collected in two different seasons: early spring, with high rainfall and high temperatures (October), and late autumn, with low rainfall and lower temperatures (May). Further analysis was carried out using ^1^H-NMR based metabolomics to analyse and compare the chemical profiles of the plants in both seasons and locations. Plant materials were collected from two sites for each season, at Wakefield farm (KwaZulu-Natal), representing a colder, wetter environment, and Telperion (Mpumalanga), representing a drier and warmer environment. Leaves of *H. aureonitens* were tested against bacteria (*Proteus vulgaris* (*P. vulgaris*) and *Bacillus subtilis* (*B. subtilis*)) as well as fungi (*Aspergillus flavus* (*A. flavus*), *Aspergillus nomius* (*A. nomius*), *Cladosporium cladosporioides* (*C. cladosporioides*), *Fusarium oxysporum* (*F. oxysporum*) and *Penicillum halotolerans* (*P. halotolerans*)). Extracts from the October harvest showed significant activities against the Gram-negative bacterium *P. vulgaris* compared to the May harvest, with an MIC value of 62.5 µg/mL. Similar activity was observed between the extracts from the wet season across the two geographically different locations. There was generally very good antifungal activity observed for all the species, with the exception of *A. nomius*, which had MIC values ranging from 0.39–1.56 µg/mL. Extracts of plant materials harvested in the wetter region had a significantly higher activity against *A. flavus* and *F. oxysporum* in both seasons than those from plants harvested in the drier region. Telperion-harvested plants exhibited better activity against *F. oxysporum* in the autumn. Hydrogen-1 NMR metabolomic analysis confirmed the significant effects of the seasons and the peculiar climates of different localities on the secondary metabolite profile of *H. aureonitens.*

## 1. Introduction

A substantial number of the African population depend on medicinal plants as their primary healthcare source. In South Africa, over 60% of the rural black population still rely on traditional healers to treat their health problems, with medicinal plant infusions (herbal teas) and concoctions administered by traditional healers rather than, or in addition to, mainstream synthetic pharmaceuticals [1]. Over 600 species of the genus *Helichrysum* are present in South Africa, Turkey, Madagascar, Eurasia and Australasia, and are reputable for their use in traditional medicine wherever they occur. For example, colds and coughs are treated with various concoctions of *H. odoratissimum*, *H. cymosum* and *H. kraussii*, while *H. nudifolium* leaves are used as a treatment for wounds and against respiratory infections [2]. The plant *H. aureonitens* Sch. Bip. (Asteraceae) is also reported by oral tradition to have a wide usage against a variety of diseases, such as enuresis and skin infection [3].

Several studies have reported the antimicrobial activities of *Helichrysum* extracts [4,5,6,7,8]. Results for the antibacterial activities of *Helichrysum* from various extractants have been reported that show that the species is more active against Gram-positive bacteria. Acetone extracts from four *Helichrysum* species, including *H. hypoleucum*, *H. odoratissimum* and *H. rugulosum*, were found to significantly inhibit the growth of five Gram-positive bacterial species, namely, *Bacillus cereus*, *B. pumilus*, *B. subtilis*, *Micrococcus kristinae* and *Staphylococcus aureus*, while none of the extracts showed any activity against the Gram-negative bacteria *E. coli*, *Klebsiella pneumoniae*, *P. aeruginosa* and *Serratia marcescens* [9]. Seven species of *Helichrysum*, including *H. araxinum*, *H. armenium*, *H. arenarium*, *H. pallasii*, *H. stoechas*, *H. sanguineum* and *H. graveolens*, were found to show antifungal activities [10].

Climate is an important seasonal factor that has a direct impact on plant ecosystem processes and structures, such as photosynthesis, nutrient cycling and transpiration, together with the production of both primary and secondary metabolites [11]. Plants are exposed to a variety of temperature levels (including extremes) when the seasons change, apart from the global climatic changes resulting in warmer and drier climates. These have an impact on their phytochemical compositions, and volatile chemicals are reported to be the compounds most affected [12]. Attempts to determine the effects that environmental changes have on plants’ phytochemical compositions provide useful information about the impact of climatic changes on the chemical profiles of plants. They are also useful with regard to determining the best time of the year or season to harvest particular plant species for the obtainment of maximum active component concentrations [13]. The literature is replete with studies on the effects of seasonal variation and climatic factors on the biological activities of plants; however, no information is available regarding the effects of environmental factors on both the biological activity and chemical properties of *H. aureonitens*. This study therefore investigated the effects of seasonal variation and different geographical locations on the biological activity of *H. aureonitens.* The test organisms were two bacterial species—*P. vulgaris* and *B. subtilis*—and five pathogenic fungi—*A. flavus*, *A. nomius*, *C. cladosporioides*, *F. oxysporum* and *P. halotolerans*. Metabolomic assessment was also conducted on the plants from the different collection sites to observe the differences in their chemical profiles relative to the growing locations and seasons.

## 2. Results

### 2.1. Antibacterial Activity of H. aureonitens Extracts

Moderate antibacterial activity was observed against *P. vulgaris*, although no activity was found at the highest concentration tested for *B. subtilis* (Table 1).

### 2.2. Antifungal Activity of H. aureonitens Extracts

In general, very good antifungal activity was observed, with values ranging from 0.39–1.56 µg/mL, except for *A. nomius*, for which no activity was observed at the highest concentrations tested (Table 2).

### 2.3. Rainfall and Temperature Data for Collection Locations

Both rainfall and temperature data were supplied by the South African weather services. The closest station to Wakefield farm from which data were acquired is in Cedara. About 15 km distant, it was assumed to have similar weather conditions to Wakefield farm. The closest station to Telperion from which data were acquired is in Witbank. The following table shows the average minimum and maximum temperatures between August 2017 and June 2018, as well as the average daily rainfall data across the two locations.

### 2.4. Metabolomics

Significant metabolic variation was observed in the study by comparing the ^1^H-NMR metabolomics data. The figures below compare the data generated from the analysis at two separate levels for ease of reporting: (i) to determine the effects of seasonal variation and different growing locations on the chemical profile of *H. aureonitens* collected from Telperion and Wakefield during the spring and autumn seasons; and (ii) to determine the effect of seasonal variation (spring and autumn) on the chemical profile of *H. aureonitens* specimens collected in wet and dry sites from both Telperion and Wakefield.

#### 2.4.1. Chemical Profile of *H. aureonitens* Collected from Telperion and Wakefield during Spring and Autumn Seasons

A total of 57 hydroalcoholic extract samples from *H. aureonitens* leaves and stems collected during spring and autumn from two geographically separate locations (Telperion and Wakefield) were subjected to ^1^H-NMR-based metabolomics to investigate how seasonal variation and different growing locations influence the plant’s chemical profile. The maximum and minimum recorded temperatures for the first collection in spring (October) in both locations were 22.3/9 °C and 24.5/10.7 °C for Wakefield and Telperion, respectively. The total monthly rainfall for the locations in October was 110 mm and 83.8 mm for Wakefield and Telperion, respectively (Table 3). The maximum and minimum recorded temperatures for the second collection in autumn (May) in both locations were 21/6.7 °C and 21.4/7.0 °C for Wakefield and Telperion, respectively. The total monthly rainfall for both locations in October was 31.8 mm and 24.6 mm for Wakefield and Telperion, respectively (Table 3). PCA and OPLS-DA models were used to analyse the data to help determine whether sufficient differences could be observed in the metabolic fingerprints of the extracts to distinguish between seasons and locations. A single sample is represented by each point on the PCA and/or OPLS-DA score scatter plot; samples having similar chemical profiles cluster together. The PCA scatter plot of all the samples across the two seasons (spring and autumn) and different locations showed slight clustering for the two seasons, but no clustering for each locality per season, as seen in Figure 1, which is indicative of greater variation in the chemical profile per season than per collection site.

An OPLS-DA model was further constructed to facilitate the clustering and identification of metabolites relevant to the determination of differences between seasons and geographical locations (Figure 2A). A clear separation was then observed between samples across seasons and locations. The model displayed a goodness of fit and predictability as presented by an R^2^X of 0.905, an R^2^Y of 0.727 and a Q^2^ of 0.384, respectively. A response permutation test (with *n* = 100) was constructed in order to validate the predictive capability of the computed OPLS-DA models. This statistical test compares the R^2^ and Q^2^ values of the true model with those of the permutated model. The test is carried out by randomly assigning the two groups and then fitting the OPLS-DA models to each permuted class variable. The permutated models’ R^2^ and Q^2^ values are then calculated and compared to the true models’ values. The results show that the measured models have significantly higher R^2^ and Q^2^ values (Figure 2B), implying that the calculated true OPLS-DA models for each dataset are statistically better relative to the 100 permuted models.

Figure 3, Figure 4, Figure 5 and Figure 6 show representative ^1^H-NMR spectra profiles for *H. aureonitens* leaves collected from wet and dry sites at both Telperion and Wakefield during spring and autumn compared for similarities or contrasts. The ^1^H-NMR spectra revealed varying peak signals across wet and dry sites from the two locations when compared, indicating significant changes in the chemical composition of *H. aureonitens* leaves. For Figure 3, Figure 4, Figure 5, Figure 6, Figure 7, Figure 8, Figure 9, Figure 10 and Figure 11, for each spectrum, the *x*-axis shows the chemical shift (in ppm) and the *y*-axis shows the signal intensity.

In Figure 3, the spectra from two wet sites in Wakefield showed conspicuous margins with taller peaks compared (especially in the aromatic region) to spectra from the two wet sites at Telperion, which is a drier location than Wakefield.

In Figure 4, spectra from the extracts collected from two dry sites from the two locations during the spring season are compared. The data for the dry site in Wakefield show comparatively taller peaks than those from Telperion, indicating the presence of a higher concentration of compounds in the aromatic region. In the aliphatic region, higher peaks are visible for the Telperion samples.

Figure 5 compares the spectra from the two wet sites at Telperion and Wakefield during the autumn season. The spectra do not show conspicuous differences, with all the samples showing peaks similar in height to the dry sites in spring. The spectra from site 1 (wet) at Telperion and the spectra from site 2 (wet) at Wakefield are very similar by visual inspection, with no remarkable difference between the two sites. A slight difference can, however, be observed between the spectra from site 2 (wet) at Telperion and the spectra from site 1 (wet) at Wakefield.

In Figure 6, ^1^H-NMR spectra from the two dry sites from both locations for the autumn season are presented. Similar to the observation for the wet sites at both locations in the autumn season, the spectra do not show conspicuous differences in terms of visible observation, with peaks of similar height as compared to the wet sites in autumn. The spectrum from site 3 (dry) at Wakefield only shows slightly taller peaks when compared to the equivalent dry site at Telperion.

#### 2.4.2. Chemical Profile of *H. aureonitens* Extracts Collected from Wet and Dry Sites at Both Telperion and Wakefield during the Spring and Autumn Seasons

Figure 7 shows the supervised multivariate analysis model (OPLS-DA) of aqueous hydroalcoholic extracts of *H. aureonitens* leaf samples collected from the wet and dry sites at Telperion during both spring and autumn seasons. The dry and wet sites in spring separated into different clusters, while the dry and wet sites show less variation in the autumn collection, with samples clustering together.

Figure 8 shows representative ^1^H-NMR spectra of *H. aureonitens* plant extracts demonstrating the chemical profile of wet sites compared to the dry site for the spring season at Telperion. The peaks in the ^1^H-NMR spectra are remarkably different by visual observation. The peaks in the aromatic region (6.5–8.0 ppm) of the wet site are much more pronounced as compared to the spectra of the extracts from the dry site, which are almost not detectable. In contrast, the peaks in the aliphatic region are more pronounced in the samples from the drier sites.

Figure 9 shows the ^1^H-NMR spectra of the leaves collected from the wet and dry sites at Telperion during the autumn season. A visual observation of the stacked spectra shows that the peaks in all the samples are very similar for both the aromatic and aliphatic regions.

Figure 10 shows representative ^1^H-NMR spectra of *H. aureonitens* plant extracts demonstrating the chemical profile of wet sites compared to the dry site for the spring season in Wakefield. The peaks in the aromatic region (6.5–8.0 ppm) of site 1 (wet) are significantly different from the peaks of site 4 (dry) by visual observation. The peaks for site 2 (wet but not as wet as site 1) are slightly similar to the peaks for site 3 (dry, but not as dry as site 4). The two sites with contrasting moisture, i.e., sites 1 and 4, show a noteworthy difference, indicating the presence of higher concentrations of bioactive compounds in the wet site.

Figure 11 shows representative ^1^H-NMR spectra of *H. aureonitens* plant extracts demonstrating the chemical profile of wet sites compared to the dry site for the autumn season for Wakefield. The peaks from all four spectra from all the sites are similar to the observation at the Telperion location during the autumn season.

The aromatic region showed significant differences between the heights of the peaks for the wet and dry sites in spring. This area was therefore considered for the annotation of compounds, which are presented in Table 4. The annotations were made by comparing the compound peaks of each compound with the representative compound peaks in the Chenomx profiler and the Human Metabolome Database (HMDB). The annotations of the compounds and their representative compound peaks are presented in Figure 12.

Figure 12 shows the peaks of the annotated metabolites in Table 4. Representative peaks of galangin, chlorogenic acid, kaempferol and quercetin are indicated in different colours.

## 3. Discussion

### 3.1. Antibacterial Activity

Hydroalcoholic extracts from the leaves and stems of *H. aureonitens* showed activity against the Gram-negative bacterium *P. vulgaris* varying from 62.5 to 250 µg/mL. However, the extracts showed no activity against the Gram-positive bacterium *B. subtilis* at the tested concentrations (Table 1). The results also showed that the extracts have better activity during the wetter season (spring) in all the sites and at both locations, with 62.5 µg/mL.

There was a clear difference noticed in the activities of extracts between the two locations during late autumn. The extracts from the wetter location (Wakefield) exhibited better activity at 125 µg/mL, with inhibition against the Gram-negative bacterium *P. vulgaris* in comparison to the drier location (Telperion), at 250 µg/mL. Previous studies have proved that the production and accumulation of primary and secondary metabolites fluctuate significantly between specimens of the same plant species grown in different environments [16,17,18].

There was also no significant difference noticed between the activities of the leaves and the stems in both locations and during the two seasons, with leaf activity at 62.5 µg/mL and stem activity at 62.5 µg/mL at both locations during spring and with 250 µg/mL activity for both leaves and stems at Telperion during autumn, contrasting with 125 µg/mL for both leaf and stem activities at Wakefield during autumn.

The antibacterial activity results obtained from *H. aureonitens* demonstrated more potency against *P. vulgaris*, which is a Gram-negative bacterium (62.5 and 125 μg/mL), than *B. subtilis*, a Gram-positive bacterium (>250 µg/mL). The result is, however, consistent with the view that most *Bacillus* species are in general less resistant [19]. Given that Gram-negative organisms are more resistant to antimicrobial compounds from plant sources than Gram-positive organisms [20], the inhibition of the growth of *P. vulgaris* at a relatively low MIC concentration between 62.5 and 125 μg/mL for the plant species tested is noteworthy and might indicate the differentiation of antibacterial compounds at the different sites that increase with water availability.

### 3.2. Antifungal Activity

With the exception of *A. nomius*, all of the fungal species were inhibited by various extract concentrations of *H. aureonitens*, with MIC values ranging generally from 0.39 to 1.56 µg/mL. Plant extracts from spring and autumn showed the same activity for *A. flavus* and *C. cladosporioides* for both seasons, although the MIC values for *A. flavus* were better for Telperion than for Wakefield. With regard to activity against *F. oxysporum*, however, there was significant activity demonstrated by the extracts of plants harvested during the wet season (spring), with 0.78 µg/mL for Telperion and 3.125 µg/mL for Wakefield, as opposed to autumn, with 3.125 µg/mL for Telperion but better activity at 1.56 µg/mL for Wakefield. The drier location in autumn, again, showed lower activity at 6.25 µg/mL as compared to the wetter locations. Plants harvested during the autumn season (the comparatively drier season) exhibited better activity at 1.56 µg/mL and 3.125 µg/mL at Telperion and Wakefield, respectively, against *P. halotolerans* compared to plants harvested during the spring season, with activity of 6.25 µg/mL and 3.125 µg/mL at Telperion and Wakefield, respectively.

A closer look at extracts used against each fungal species from the two locations exhibited differing activities. Extracts from both locations maintained almost the same activities against *A. flavus*. Activity against *C. cladosporioides* was the same for each location in spring, at 0.78 µg/mL, just like that against *A. flavus*, at 0.39 µg/mL at Telperion and in the range of 0.78 to 1.56 µg/mL at Wakefield, except there was no observed significant difference in activity between each of the sites, as activities at the sites were around 0.78 µg/mL.

For the spring season, there was a better activity at the drier location (Telperion) than Wakefield (wetter) against *F. oxysporum*, at 0.78 g/mL and 3.125 g/mL, respectively. Extracts harvested from the autumn season, however, demonstrated activity that was the opposite of what was observed for spring against *F. oxysporum* for the two locations. Better activities of extracts from Wakefield (wetter region) were, however, exhibited against *F. oxysporum*, with values of 1.56 g/mL for Telperion (drier region) and at 3.125 g/mL during autumn.

The acetone extract of *H. aureonitens* showed variable activities against the tested fungi. Based on reports of past studies on *Helichrysum* species and a handful of medicinal plants, acetone extracts showed considerable activity against the tested fungi, while rather low activities by comparison were recorded against the same fungi when hydroalcoholic solvents were used [6,9,21,22]. The use of acetone as a solvent, in contrast to hydroalcoholic extracts used in the other analysis, also probably explain the misalignment of the antifungal activity with the metabolomic analysis. Chlorogenic acids have been identified in various *Helichrysum* species, especially in *H. aueonitens*, often linked to the biological activity observed in the plants [23,24,25]. Chlorogenic acids are soluble in a wide variety of solvents ranging from water to acetone [26,27,28], whereas galangin, quercetin and kaemfrol are less soluble in polar solvents. Acetone would therefore additionally extract compounds such as galangin, kaempferol and quercetin, therefore supporting the differential activity found in the hydroalcoholic and acetone extracts.

In the antifungal assays, there was no clear distinction in terms of better activity for Wakefield or the wetter sites, as variable results were obtained for all the fungal species. However, in general, very good activity, ranging from 0.39–1.56 µg/mL, was achieved against the fungal species with acetone as the extractant, indicating that *H. aureonitens* contains strong antifungal compounds. Since no study has reported the effects of seasonal variation on the antifungal properties of *Helichrysum* species, the influence of seasonality on the antifungal properties of other medicinal plants were compared. In support of the findings of this study, de Macedo et al. [29] reported that, based on their study on the effect of seasonality on the antifungal properties of *Psidium salutare*, a medicinal plant native to Brazil, they could not establish the influence of any of the three seasonal collection periods on the antifungal properties of the plant. Similarly, no effect of season on the antifungal activities of seaweeds was observed by Stirk et al. [24] in their study on seasonal variation in the antifungal activities of seven seaweeds from South Africa.

### 3.3. Metabolomics

Due to the economic importance and the many medicinal uses of *H. aureonitens*, determining the effect of varying climatic conditions provides insights into the harvesting periods/conditions or seasons that will yield the highest amounts of phytochemicals and the best possible geographical locations, which information is critical to the maximization of its medicinal potential. There is, however, a scarcity of data comparing seasonal metabolite changes in *H. aureonitens*. This is also important given that climate change is generally creating drier and warmer locations, which might impact significantly the medicinal and biological activities of plants used for medicinal purposes.

A PCA was performed to determine whether there was an observable clustering pattern. The PCA showed loose clustering of the samples based on season (Figure 1). Further analysis was carried out using the supervised OPLS-DA model to obtain clear seasonal class groupings between samples across seasons and collection locations (Figure 2A). Figure 3 showed that the wet collections sites in spring at Telperion had much lower concentrations of the aromatics than the wet collection sites of Wakefield in the spring. Much lower concentrations of aromatics were observed for the dry sites in spring at both locations, although the concentrations of aromatics were significantly lower for Telperion (Figure 4). This could be explained by the slightly higher rainfall in August–October and the lower temperature for Wakefield compared to Telperion (Table 4). When spectra from the wet sites in both Telperion and Wakefield during the autumn season were compared, no conspicuous margin was seen (Figure 5), indicating that the lower rainfall and similar lower temperatures resulted in lower concentrations of aromatic compounds at both sites (Table 4). The heights of the peaks of the aromatic compounds were similar to those for the dry conditions at Wakefield in spring.

Figure 7 clearly shows the separation of the dry location from the wetter locations in spring at Telperion. However, this was not observed for the autumn collection, as the dry site clustered with the wetter location, indicating a change in chemical profile as a result of water availability.

At Telperion in spring, the two wet collection sites showed much higher concentrations of aromatics than the dry collection site (Figure 8). This observation was the same for the dry collection site in autumn, which also showed similar lower concentrations of aromatics (Figure 9), thereby supporting the clustering observed in Figure 7. This, again, supports the notion that higher rainfall might be conducive to aromatic production in *H. aureonitens*.

Similarly, the dry collection sites at Wakefield showed much lower concentrations of the aromatics than the wet collection site in spring (Figure 10). Once again, low concentrations of aromatics for the wet and dry sites at Wakefield during autumn, as shown in Figure 11, are congruent with the observation at Telperion, where low concentrations of aromatics were also observed at both wet and dry sites.

Several studies have linked increased rainfall to increase in the production of chlorogenic acid [30,31]. In a recent study on the effects of increase in atmospheric CO_2_ and other climatic variations on phenolics in coffee trees, [32] reported that phenolic levels were positively correlated with the rainy season. The report further showed that chlorogenic acid concentration, in particular, was reduced during the dry season. In another study conducted on walnut leaf samples by [33], the authors reported a phenomenal increase in phenolic compounds, which included 3-caffeoylquinic acid, when walnut leaf samples from nine different cultivars were investigated for their phenolic compounds. The considerable changes observed between three consecutive production years were attributed to climatic factors, mainly temperature and rainfall. Samples harvested from the year with the highest amount of rainfall (2002) showed a surge in the production of caffeoylquinic acid and other phenolic compounds.

The antibacterial assay results align very well with what has been found in the metabolomics analysis, with higher activity obtained where an increase in aromatic compounds was observed. Numerous studies have reported the antibacterial activities of compounds such as quercetin, kaempferol and chlorogenic acid [34,35,36], supporting the increased levels of aromatics in the metabolomics analysis. The antifungal results however, are highly variable, possibly because of the use of acetone as an extractant. Even though the results do not align with the metabolomics analysis, using a proven solvent for antifungal analysis resulted in excellent antifungal activity. It is therefore expected that better alignment with the antifungal results would be obtained using similar solvents.

## 4. Materials and Methods

### 4.1. Plant Material Collection

Whole plant materials of *H. aureonitens* were collected at two climatically different sites at two different seasons of the year, namely, spring (late October) and autumn (early May). The experiment was designed with the aim of comparing a “treatment” group with a “control” group in both locations. The wet sites were chosen as the “control” groups, while the dry sites were considered the “treatments” in each of the locations in the experiment. Telperion Nature Reserve is situated in Mpumalanga (25.7039° S, 28.9814° E) and Wakefield farm in the KwaZulu-Natal Midland region (29°30′0″ S and 29°54′0″), representing a warmer, drier climate and a cooler, wetter climate, respectively. Three batches of representative plant samples were collected at different sites for each location, transferred into brown paper bags (10 cm × 20 cm) and transported to the laboratory. Plant materials were identified, and representative voucher specimens with the names WAHA-01, WAHA-02, WAHA-03 and WAHA-04 were deposited in the UNISA Science Campus horticulture centre’s herbarium.

*Helichrysum aureonitens* have very tough fibrous tissues and therefore requires a longer time to completely dry. Since one of the main objectives of the study was to determine the chemical profiles of the plants, any method that would interfere with the drying process and thereby cause a loss or reduction in bioactive compounds was avoided. In a review that investigated the efficiency of drying conditions for essential oil production from aromatic plants, Özgüven et al. [37] found that better results were obtained with natural drying methods. Mirhosseini et al. [38] similarly discovered that, due to the lower temperatures employed in the shade-drying of *Stachys lavandulifolia* (an important plant consumed as a herbal tea in Iranian folk medicine), as the evaporation of fragrant composition is comparatively lower than in both oven- and sun-drying, the amounts of essential oils were found to be greater in shade-dried samples. In view of this, the plant samples in this study were shade-dried at the ambient temperature of the laboratory and monitored weekly until they were properly dried by the fourth month. Furthermore, the stems and leaves of each plant were separated and stored in transparent cellophane bags at an ambient temperature until further analysis.

### 4.2. Preparation of Plant Extracts and Antimicrobial Testing

#### 4.2.1. Preparation of Plant Extracts

Plant materials for the antibacterial assay were ground and pulverized into powder form using a kitchen blender and a pestle and mortar, and were extracted using methanol–water (70:30), according to Nadia et al. [39]. The mixtures were shaken for 5 min in a shaker (Thermo Fisher Scientific, United States), centrifuged at 3000 rpm for 10 min (Eppendorf 5424, Saxony, Germany) and further decanted and filtered by discarding the plant remainders. Extracts were concentrated to dryness by evaporation in a drying chamber (Airvolution dryer, South Africa), yield was determined and they were stored in a refrigerator at 4 °C until further use.

Plant materials for the antifungal assay were prepared according to the above description. A ratio of 10:1 acetone-to-plant materials (50 mg of plant material to 500 mL of acetone) was used for the extraction in a centrifuge tube, according to Eloff [40]. Samples were put in a shaker overnight at 130 rpm (Thermo Fisher Scientific, Waltham, MA, USA), following which samples were centrifuged at 3000 rpm for 10 min (Eppendorf microcentrifuge 5427R, Saxony, Germany) and the supernatant was transferred to a glass vial. A stream of air at room temperature was used to remove the remaining solvent from the extracts already placed in pre-weighed glass tubes and stored at room temperature.

#### 4.2.2. Antibacterial Activity

The Gram-positive bacterium *B. subtilis* (ATCC 33420) and the Gram-negative bacterium *P. vulgaris* (ATCC 84270) (Anatech, KwikStick^®^) were used for this study. The organisms were maintained in nutrient agar slants (Biolab) and later recovered for testing by growing them in a nutrient broth slant (Sigma-Aldrich, Burlington, MA, USA).

The experiment was performed in duplicate. Three biological replicates were used to determine the initial MIC. Thereafter, the experiment was repeated and the MIC confirmed [41]. The MIC was determined using the 96-well plate microdilution method at the tested concentrations: 250, 125, 62.5 and 31.25 µg/mL. Bacteria with nutrient agar and extract was used as a negative control, while gentamicin (1000 µg/mL) (Sigma-Aldrich, Burlington, MA, USA) was used as a positive control. The extracts were then dried and redissolved in 30% acetone (used as the vehicle control). This step was included in the experiment to ensure that the observed activity against the bacteria did not originate from the applied solvent. Plates were incubated with natural air circulation at 37 °C (Memmert incubator IN55, USA). Afterwards, 40 μL of 2 mg/mL p-iodonitrotetrazolium (INT) chloride was added to each well (Sigma-Aldrich, Schnelldorf, Germany) to determine the minimum inhibitory concentration (MIC), where a change in colour to pink indicated bacterial growth.

#### 4.2.3. Antifungal Activity

The fungal isolates used in the study were *A. flavus* (MRC 3951), *A. nomius* (PPRI 3753), *C. cladosporioides* (PPRI 10367), *F. oxysporum* (MRC 1907) and *P. halotolerans* (PPRI 25804). The isolates were maintained on potato dextrose agar (PDA) and stored at 4 °C. The fungal cultures were subcultured into plates of newly prepared potato dextrose broth (PDB) growth medium from potato dextrose agar (PDA) slants and incubated at 30 °C for 3 days. A full loop of each actively growing fungal species was introduced into 50 mL freshly prepared PDB and reincubated for another 3–4 days until some turbidities were observed, indicating active growth. 

The broth microdilution method was used to determine the antifungal activity of the plant extracts. For this, 100 μL of plant extracts were introduced into the first row of the 96-well plates, followed by another 100 μL of autoclaved distilled water. A total of 200 μL in each first well ‘A’ was serially diluted until well ‘H’, followed by 100 μL aliquot of each fungal species inoculum. Positive and negative controls were prepared with amphotericin B (1 mg/mL) and 30% acetone, respectively. Afterwards, 40 μL of 2 mg/mL INT chloride was added to each well (Sigma-Aldrich, Schnelldorf, Germany). The plates were sealed with parafilm and incubated for 24 and 48 h at 30 °C, with 100% humidity. Three biological replicates were used to determine the initial MIC. Thereafter, the experiment was repeated and the MIC confirmed [42]. The lowest concentration of plant extract that suppressed fungal growth as observed by colour change after 48 h of incubation was established to be the minimum inhibitory concentration (MIC) and thus recorded.

### 4.3. Metabolomics Analysis

Fifty milligrams of powdered leaf material was weighed and stored in 2 mL Eppendorf tubes and extracted following an established direct extraction method. Plant material was extracted with 0.75 mL deuterated methanol (CD_3_OD) and 0.75 mL of deuterium water (D_2_O) (pH 6.38) with potassium dihydrogen phosphate (KH_2_PO_4_) and 0.1 % (*w*/*w*) TSP (Trimethylsilylpropionic acid sodium salt) was added. The samples were vortexed for 1 min at room temperature to combine the reagents. After ultrasonically (Branson 2800, SonicsOnline, USA) breaking down the cell walls for 15 min, the mixtures were centrifuged for 20 min to separate the supernatants from the pellets. Each tube’s supernatant was then transferred to a 5 mm NMR tube for analysis on a 600 MHz NMR spectrometer (Varian Inc., Palo Alto, CA, USA) with 32 scans.

Spectral data obtained from the NMR were pre-processed with MestReNova software (9.0.1, Mestrelab Research, Santiago, Spain) and subjected to phase correction, baseline correction, referencing and normalization. TSP was included as a standard in all samples and quality control of the analysis was performed by confirming five sharp peaks for methanol. TSP was also used to reference and normalise all samples. Additionally, MestReNova was also used for bucketing NMR spectra. The spectral intensities were then reduced to integrated sections of identical width (0.04 ppm each) corresponding to the 0.04–10.00 ppm range in a process often referred to as binning. Afterwards, the resulting ASCII files were imported into Microsoft Excel 2013. SIMCA-P software (version 13.0, Umetrics, Umea, Sweden) was used to conduct the PCA and OPLS-DA multivariate data analyses, while further transformation of the data was carried out with Pareto scaling. The final data do not include the residual water peak (4.60–5.00 ppm) and the methanol peak (3.28–3.36 ppm) [43].

### 4.4. Compound Annotation

The Chenomx NMR suite and the Human Metabolome Database were used to detect metabolites, employing TSP as a reference and the integrated metabolite spectrum libraries. Data from the existing literature were used to verify the annotated chemicals.

### 4.5. Rainfall and Temperature Data

Both rainfall and temperature data were supplied by the South African weather services. The closest station to Wakefield farm from which data were obtained was in Cedara. At a distance of about 15 km, it was assumed to have similar weather conditions to Wakefield farm. The closest station to Telperion from which data were obtained was in Witbank. Table 3 shows the average minimum and maximum temperatures between August 2017 and June 2018, as well as the average daily rainfall data across the two locations.

## 5. Conclusions

This study reports for the first time a comparison of the antibacterial and antifungal activities of samples of a *Helichrysum* species collected in different climatic regions and different seasons supported by metabolomic analysis. The study shows that plant materials collected in spring (higher rainfall) show better activity than samples collected in autumn for wet sites, as the dry sites showed comparably lower activity in spring and autumn. The acetone extracts showed generally very good antifungal activity at 0.39–1.56 µg/mL, although significant variation was observed, with no activity shown by any sample against *A. nomius* and *A. flavus*, and similar activity shown against *C. cladosporioides* for the spring and autumn locations for the sites, although better activity was shown for Telperion. Only *P. halotolerans* showed better activity in autumn compared to the spring samples for both sites. In contrast to the antibacterial activity, the data for which are supported by the metabolomic analysis, seasonal variation showed very little or no impact on antifungal activity, which could be attributed to the use of a different solvent. Furthermore, an understanding of the distribution of secondary metabolites as influenced by seasons and different growing locations in *H. aureonitens* in this study adds to our understanding of the phytochemistry of this plant. A comparison between the aromatic contents of *H. aureonitens* growing in two geographically diverse locations also showed that levels of aromatics are favoured in the wetter sites for both geographical locations in spring. The study therefore concludes that aromatics is positively linked with the rainy season and lower temperatures in *H. aureonitens.* Compounds such as chlorogenic acid, quercetin and kaempferol are compounds with numerous reports on their biological activity and are therefore important for medicinal preparations. Information on the environmental conditions that are different for various collection times and locations are therefore important, and changes towards warmer and drier climates will decrease aromatic compound production in *H. aureonitens*.

## Figures and Tables

**Figure 1 metabolites-12-00758-f001:**
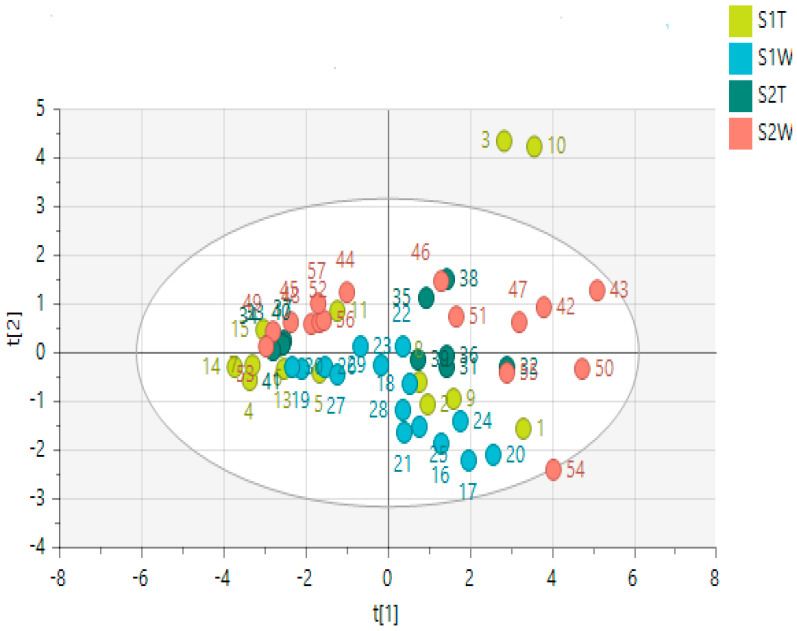
PCA score scatter plot showing component 1, *x*−axis (PC1 = 54.6%), and component 2, *y*−axis (PC2 = 14.6%), of *H. aureonitens* leaves and stems collected during spring and autumn at Telperion and Wakefield. S1T = spring, Telperion; S1W = spring, Wakefield; S2T = autumn, Telperion; S2W = autumn, Wakefield.

**Figure 2 metabolites-12-00758-f002:**
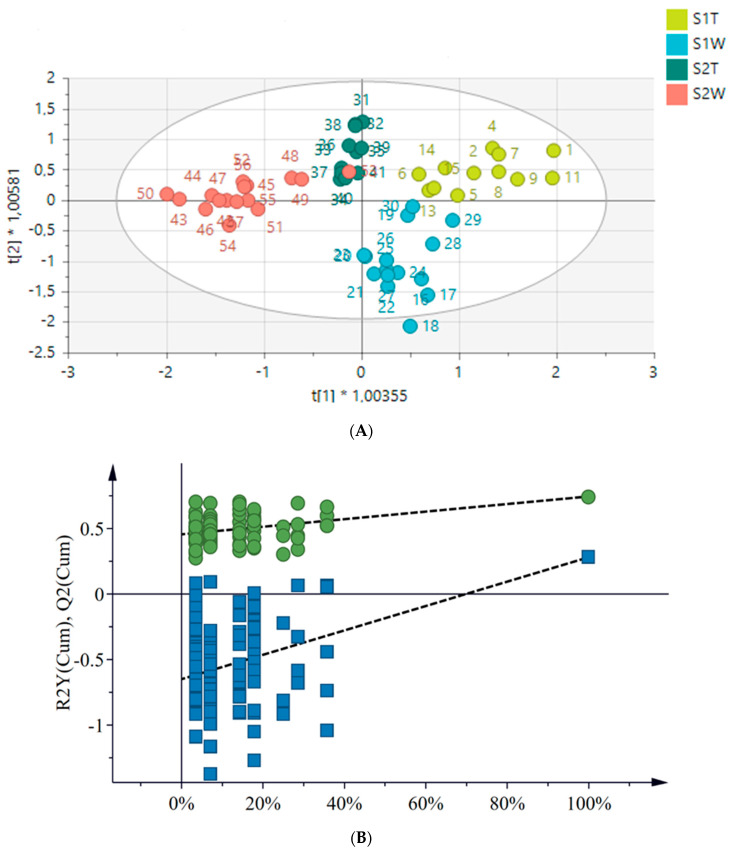
(**A**) An OPLS-DA score scatter plot showing the predictive (*x*−axis) and orthogonal (*y*−axis) components of *H. aureonitens* leaves and stems during spring and autumn at Telperion and Wakefield. R^2^X = 0.905, R^2^Y = 0.727 and Q^2^ = 0.384. S1T = spring, Telperion; S1W = spring, Wakefield; S2T = autumn, Telperion; S2W = autumn, Wakefield. (**B**) The response permutation test (*n* = 100) for the OPLS-DA model corresponding to *y*-axis intercepts. R^2^ = (0.0, 0.46) and Y^2^ = (0.0, −0.65).

**Figure 3 metabolites-12-00758-f003:**
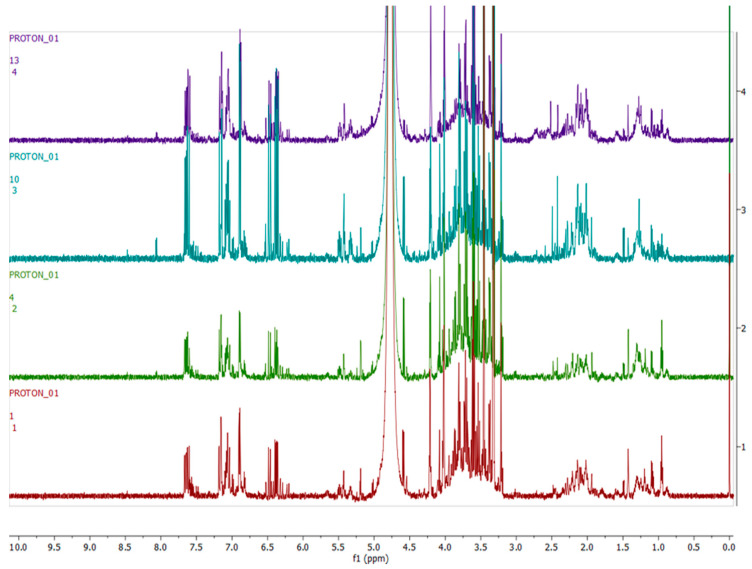
Comparison between samples harvested from wet sites in the spring season across the two locations on the basis of 600 MHz ^1^H-NMR spectra of *H. aureonitens* leaf extracts. Red—spring season, site 1 (wet) Telperion; green—spring season, site 2 (wet) Telperion; light blue—spring season, site 1 (wet) Wakefield; purple—spring season, site 2 (wet) Wakefield.

**Figure 4 metabolites-12-00758-f004:**
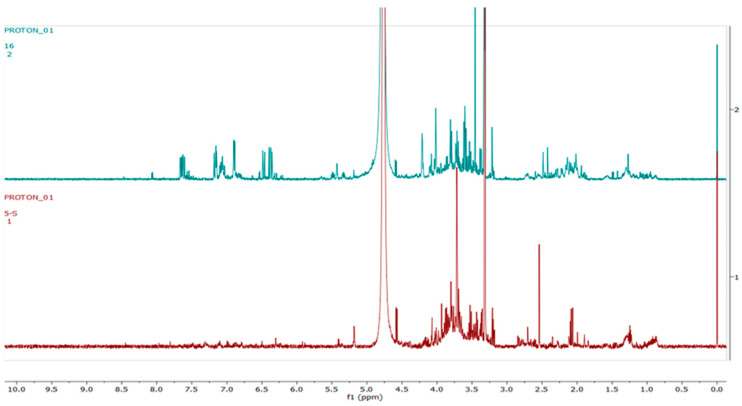
Comparison between samples harvested from dry sites in the spring season across the two locations on the basis of 600 MHz ^1^H-NMR spectra of *H. aureonitens* leaf extracts. Red—spring season, site 3 (dry) Telperion; light blue—spring season, site 3 (dry) Wakefield.

**Figure 5 metabolites-12-00758-f005:**
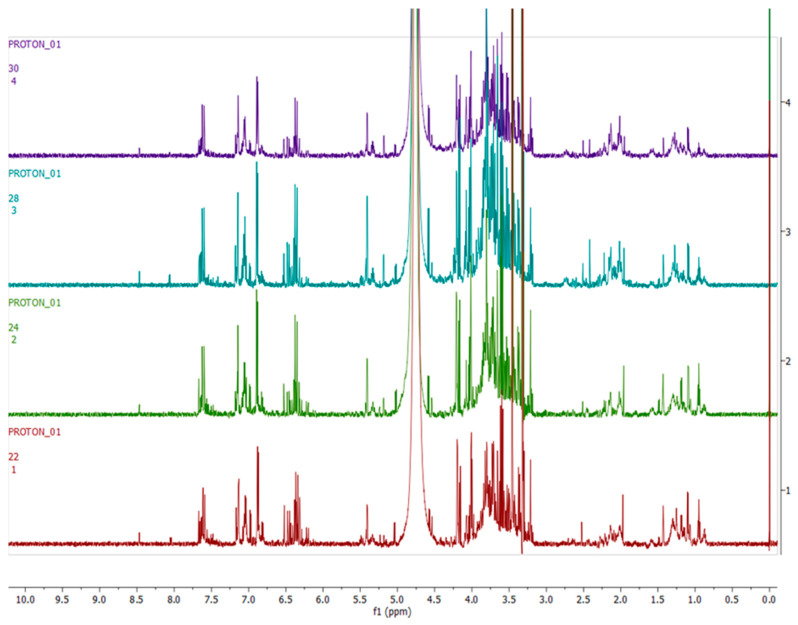
Comparison between samples harvested from wet sites in the autumn season across the two locations on the basis of 600 MHz ^1^H-NMR spectra of *H. aureonitens* leaf extracts. Red—autumn season, site 1 (wet) Telperion; green—autumn season, site 2 (wet) Telperion; light blue—autumn season, site 1 (wet) Wakefield; purple—autumn season, site 2 (wet) Wakefield.

**Figure 6 metabolites-12-00758-f006:**
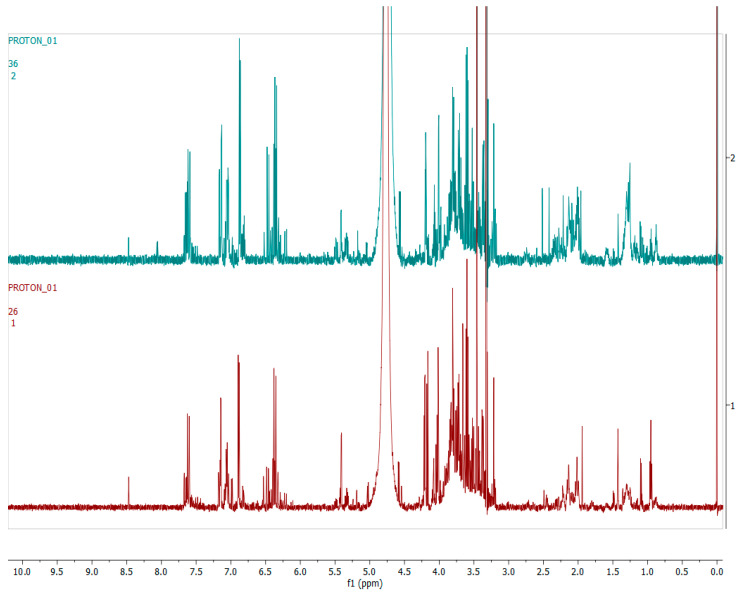
Comparison between samples harvested from dry sites in the autumn season across the two locations on the basis of 600 MHz ^1^H-NMR spectra of *H. aureonitens* leaf extracts. Red—autumn season, site 3 (dry) Telperion; light blue—autumn season, site 3 (dry) Wakefield.

**Figure 7 metabolites-12-00758-f007:**
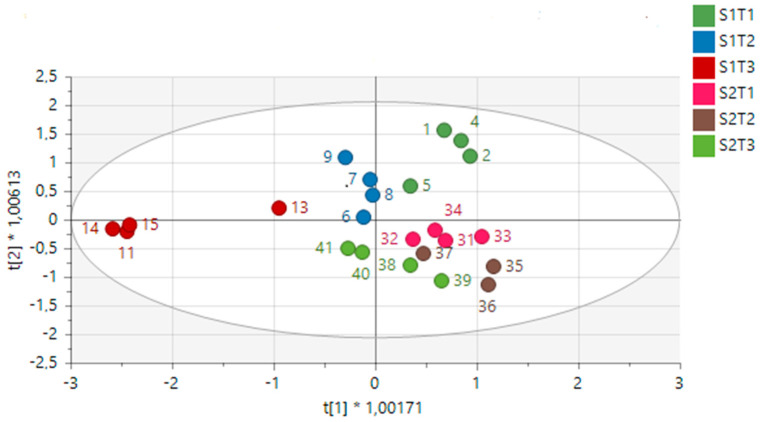
OPLS-DA score scatter plot of *H. aureonitens* leaf extracts collected from the wet and dry sites at Telperion during both spring and autumn seasons. R^2^X = 0.948, R^2^Y = 0.715 and Q^2^ = 0.024. Green (S1T1) = spring site 1(wet); blue (S1T2) = spring site 2 (wet); red (S1T3) = spring site 3 (dry); pink (S2T1) = autumn site 1 (wet); brown (S2T2) = autumn site 2 (wet); olive (S2T3) = spring site 3 (dry).

**Figure 8 metabolites-12-00758-f008:**
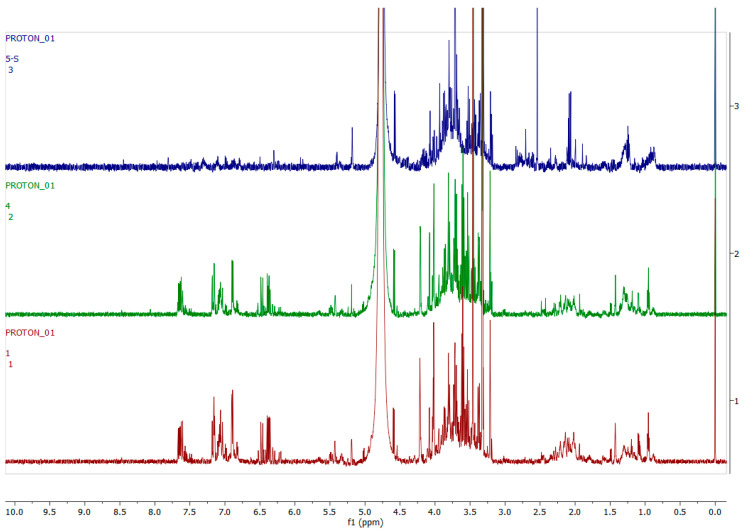
The 600 MHz ^1^H-NMR spectra of *H. aureonitens* leaf extracts collected from the wet and dry sites at Telperion during the spring season. Red = site 1 (wet); green = site 2 (wet); blue = site 3 (dry).

**Figure 9 metabolites-12-00758-f009:**
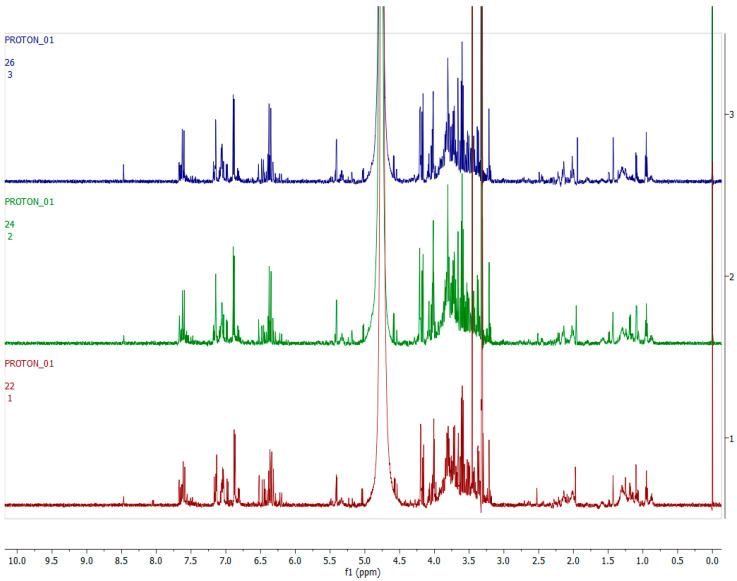
The 600 MHz ^1^H-NMR spectra of *H. aureonitens* leaf extracts collected from the wet and dry sites at Telperion during the autumn season. Red = site 1 (wet); green = site 2 (wet); blue = site 3 (dry).

**Figure 10 metabolites-12-00758-f010:**
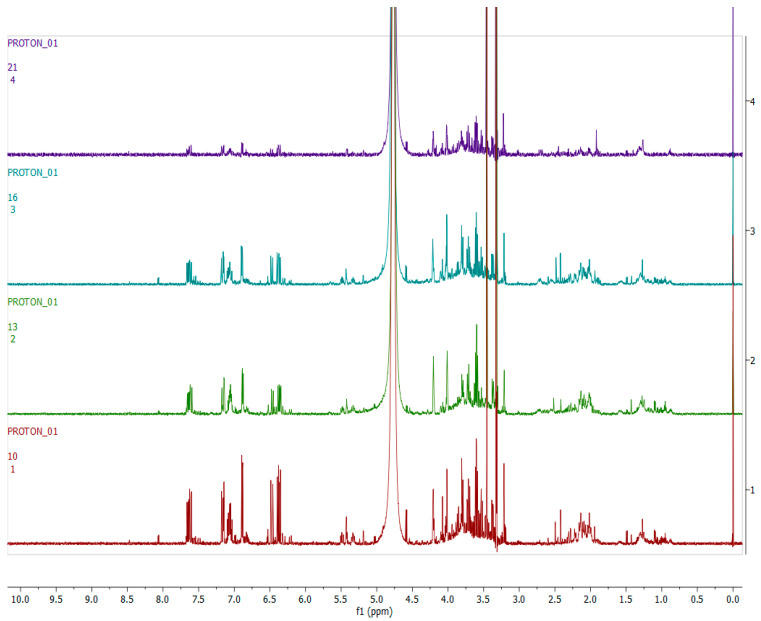
The 600 MHz ^1^H-NMR spectra of *H. aureonitens* leaf extracts collected from the wet and dry sites at Wakefield during spring season. Red =site 1 (wet); green = site 2 (wet); blue = site 3 (wet); purple = site 4 (dry).

**Figure 11 metabolites-12-00758-f011:**
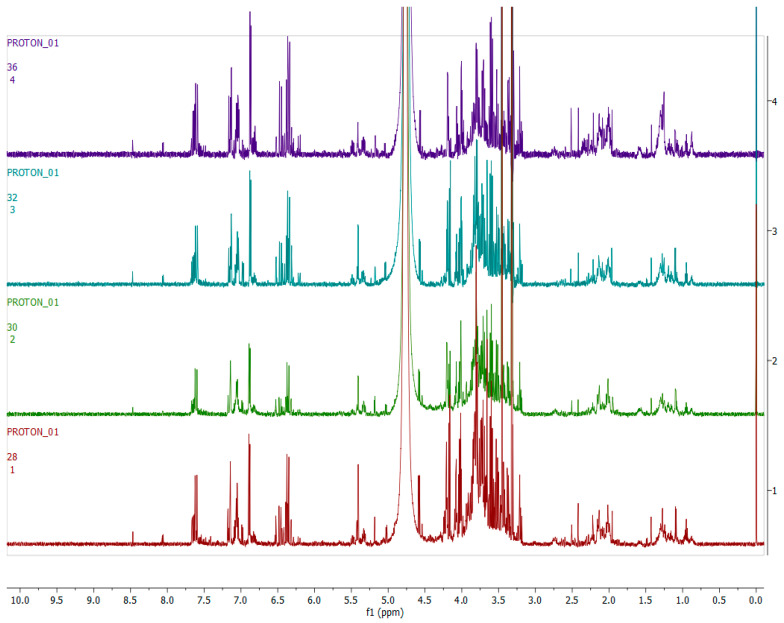
The 600 MHz ^1^H-NMR spectra of *H. aureonitens* leaf extracts collected from the wet and dry sites at Wakefield during autumn season. Red = site 1 (wet); green = site 2 (wet); blue = site 3 (dry); purple = site 4 (dry).

**Figure 12 metabolites-12-00758-f012:**
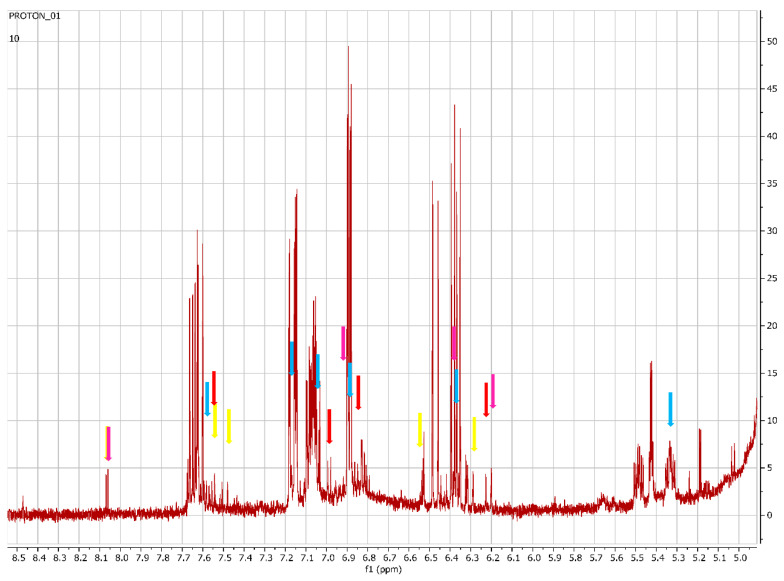
Peaks of the annotated compounds indicated for each of the compounds. Yellow = galangin (8.06, 7.58, 7.49, 6.54, 6.28 ppm); purple = kaempferol (8.06, 6.94, 6.44, 6.20 ppm); blue = chlorogenic acid (7.56, 7.06, 7.03, 6.83, 6.22, 5.33 ppm); red = quercetin (7.66, 7.54, 6.88, 6.20 ppm).

**Table 1 metabolites-12-00758-t001:** Antibacterial activity of the hydroalcoholic extracts from leaves and stems of *H. aureonitens*, with the MIC values (µg/mL) obtained from different sites at two different climatic locations and in two different seasons of the year against the Gram-negative bacterium *P. vulgaris* and the Gram-positive bacterium *B. subtilis*.

Plant Part	Season	Location	*P. vulgaris*	*B. subtilis*
Leaf (Site 1)	Spring	Telperion	62.5	>250
Stem (Site 1)	Spring	Telperion	62.5	>250
Leaf (Site 2)	Spring	Telperion	62.5	>250
Stem (Site 2)	Spring	Telperion	62.5	>250
Leaf (Site 1)	Spring	Wakefield	62.5	>250
Stem (Site 1)	Spring	Wakefield	62.5	>250
Leaf (Site 2)	Spring	Wakefield	62.5	>250
Stem (Site 2)	Spring	Wakefield	62.5	>250
Leaf (Site 1)	Autumn	Telperion	62.5	>250
Stem (Site 1)	Autumn	Telperion	62.5	>250
Leaf (Site 2)	Autumn	Telperion	62.5	>250
Stem (Site 2)	Autumn	Telperion	62.5	>250
Leaf (Site 3)	Autumn	Telperion	>250	>250
Stem (Site 3)	Autumn	Telperion	250	>250
Leaf (Site 1)	Autumn	Wakefield	250	>250
Stem (Site 1)	Autumn	Wakefield	125	>250
Leaf (Site 2)	Autumn	Wakefield	125	>250
Stem (Site 2)	Autumn	Wakefield	125	>250
Leaf (Site 3)	Autumn	Wakefield	125	>250
Stem (Site 3)	Autumn	Wakefield	125	>250
Gentamicin				1000

**Table 2 metabolites-12-00758-t002:** The MIC values (µg/mL) for extracts from different parts of *H. aureonitens* specimens obtained from different sites at two different climatic locations and in two different seasons of the year against the fungi *Aspergillus flavus* (*A. flavus*), *Aspergillus nomius* (*A. nomius*), *Cladosporium cladosporioides* (*C. clados*), *Fusarium oxysporum* (*F. oxy*) and *Penicillium halotolerans* (*P. halo*).

Plant Part	Season	Location	*A. flavus*	*A. nomius*	*C. clados*	*F. oxy*	*P. halo*
Leaf (Site 1)	Spring	Telperion	0.39	>250	0.78	0.78	6.25
Stem (Site 1)	Spring	Telperion	0.39	>250	0.78	0.78	6.25
Leaf (Site 2)	Spring	Telperion	0.39	>250	0.39	0.78	6.25
Stem (Site 2)	Spring	Telperion	0.39	>250	0.78	1.56	3.125
Leaf (Site 1)	Spring	Wakefield	0.78	>250	0.78	3.125	3.125
Stem (Site 1)	Spring	Wakefield	0.78	>250	0.78	3.125	6.25
Leaf (Site 2)	Spring	Wakefield	0.78	>250	1.56	3.125	6.25
Stem (Site 2)	Spring	Wakefield	0.78	>250	0.78	6.25	3.125
Leaf (Site 3)	Spring	Wakefield	1.56	>250	0.78	6.25	3.125
Stem (Site 3)	Spring	Wakefield	1.56	>250	0.78	3.125	3.125
Leaf (Site 4)	Spring	Wakefield	1.56	>250	0.78	3.125	3.125
Stem (Site 4)	Spring	Wakefield	1.56	>250	1.56	6.25	3.125
Leaf (Site 1)	Autumn	Telperion	0.39	>250	0.78	3.125	1.56
Stem (Site 1)	Autumn	Telperion	0.39	>250	0.78	3.125	1.56
Leaf (Site 2)	Autumn	Telperion	0.39	>250	0.78	3.125	1.56
Stem (Site 2)	Autumn	Telperion	0.39	>250	0.78	3.125	1.56
Leaf (Site 3)	Autumn	Telperion	0.39	>250	0.39	3.125	1.56
Stem (Site 3)	Autumn	Telperion	0.39	>250	0.78	6.25	1.56
Leaf (Site 1)	Autumn	Wakefield	0.78	>250	1.56	1.56	1.56
Stem (Site 1)	Autumn	Wakefield	0.78	>250	0.78	1.56	1.56
Leaf (Site 2)	Autumn	Wakefield	0.78	>250	0.39	1.56	1.56
Stem (Site 2)	Autumn	Wakefield	0.78	>250	1.56	1.56	1.56
Leaf (Site 3)	Autumn	Wakefield	1.56	>250	0.78	1.56	3.125
Stem (Site 3)	Autumn	Wakefield	1.56	>250	1.56	1.56	3.125
Leaf (Site 4)	Autumn	Wakefield	3.125	>250	1.56	6.25	3.125
Stem (Site 4)	Autumn	Wakefield	6.25	>250	1.56	6.25	3.125
Amphotericin B							1000

**Table 3 metabolites-12-00758-t003:** Average minimum and maximum temperatures and daily rainfall data between August 2017 and June 2018 for Cedara and Witbank.

	Wakefield (Cedara Data)			Telperion (Witbank Data)		
Month	Avg Daily Max T (°C)	Avg Daily Min T (°C)	Total Monthly Rainfall (mm)	Avg Daily Max T (°C)	Avg Daily Min T (°C)	Total Monthly Rainfall (mm)
August	20.8	5.2	3.6	21.3	5.6	3.8
September	24.5	8.9	9.6	26.8	9.8	27.2
October	22.3	9	110	24.5	10.7	83.8
November	23.7	10.7	105.8	27	11.9	109.2
December	23.8	12.6	79	26.5	14.1	153.2
January	27.3	14.2	79.4	27.9	13.9	71.8
February	26.5	15.3	170.2	26.3	15	75.8
March	25.7	13.7	124.2	26.0	13.4	148.4
April	23.9	12.2	52.4	23.9	12.0	26.0
May	21	6.7	31.8	21.4	7.0	24.6
June	20.2	3.4	1	20.1	4.5	0.2

**Table 4 metabolites-12-00758-t004:** The ^1^H NMR peaks (ppm) of annotated compounds that contributed to the observed separation of extracts collected from Wakefield in both seasons relative to the Telperion collections across both seasons.

Compound	^1^H-NMR Chemical Shift	Referenced ppm	Reference
Galangin	8.06, 7.58, 7.49, 6.54, 6.28	8.25, 7.49–7.57, 6.55, 6.28	[6]
Chlorogenic acid	7.56, 7.06, 7.03, 6.83, 6.22, 5.17, 3.95, 2.14, 1.97	7.66, 7.18, 7.0, 6.9, 6.4, 5.3, 3.86, 1.74–2.04	[14]
Kaempferol	8.06, 6.94, 6.44, 6.20	8.03, 6.93, 6.43, 6.18,	[15]
Quercetin	7.66, 7.54, 6.88, 6.20	7.67, 7.53, 6.88, 6.40, 6.18	[15]

## Data Availability

The data presented in this study are available in the article.

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
