# Peer review of "Effect of Different Climatic Regions and Seasonal Variation on the Antibacterial and Antifungal Activity, and Chemical Profile of *Helichrysum aureonitens* Sch. Bip"

_metabolites, 2022, doi:10.3390/metabo12080758_

Round 1
Reviewer 1 Report
This is a properly designed and conducted study that deserves publication.
Author Response
Thanks for your comments.
Reviewer 2 Report
I appreciate and acknowledge the efforts of the authors in making the corrections. However, a control of the vehicle type using the solvent used to dissolve the extracts and perform the biological tests is still lacking.
Another critical point, as cited in my previous review, and which remains, would be the metabolomics analysis of a different extract than the one used in the biological tests. This is cited even by the authors in lines 347-348 : « The use of acetone as solvent in contrast to hydroalcoholic extracts used for the other analysis also probably explain the misalignment of the antifungal activity to the metabolomic analysis. «
This makes it very complicated, not to say impossible, to compare the results. Although the authors claim a certain correlation between the hydroalcoholic extracts and the antifungal tests, these extracts are also different (one 70% MeOH and one 50% MeOH).
In my opinion, the lack of appropriate control groups and the use of different extracts are critical points that make the publication unfeasible, despite the important efforts to collect extracts in different locations and seasons.
Reviewer 3 Report
1. A major problem with the study is the writing. Although the English writing is overall correct, in many places in the manuscript the style is inappropriate for a scientific manuscript. The text should undergo a scientific editing. Some examples from the abstract:
The sentence in lines 11-14 is too long, and also is illogical. It should be revised for clarity, and be divided to shorted sentences.
The sentence in lines 20-21 is also illogical. It should be revised for clarity and precision of the English language style.
The beginning of the sentence in line 20 “Equal activity was observed with the extracts” should be revised. With is not the correct word to be used. And equal activity is not a conventional scientific terminology.
2. The sentence in lines 62-64 contain a repetitive information. It should be revised.
3. The sentence in lines 68-72 is too long.
4. Line 431 “big brown paper bags” is not a suitable scientific description.
5. Section 4.1: About the drying of the plant material. The discussion of this information should start in a new line, and not immediately following the accession numbers. Also, some information should be given about the ambient temperature in the lab during drying.
6. The title of section 4.2.1. should be revised to Antibacterial activity
7. The title of section 4.2.2. should be revised to Antifungal activity
8. in section 4.2: the first paragraph does not belong in the antibacterial sub-section. It should be written under a separate section “ Preparation of plant extracts”.
9. Another major issue with the manuscript is the lack of statistical analyses. Specifically, Table 1: The table presents the values for the 3 replicates for each site. This I s not the conventional way to present scientific data. The 3 replicates should be averaged for each location and season. And a statistical analysis should be applied to evaluate potential differences between treatments.
10. The same comment applies to Table 2. Averages and a statistical analyses should be presented rather than the values for the individual replications.
11. The authors are encouraged to find a way to present the data, or at list part of it in a figure.
12. Figure 1: the figure is blurry. It should be replaced with a figure of better resolution. Also, the figure does not have proper titles for the X and Y axes.
13. Figure 2: Both sub- figures do not have proper titles for the X and Y axes.
14. Figures 3-10: The figures do not have proper titles for the X and Y axes. Also, should there be a numerical way to quantify differences other than the peaks height and visual comparisons?
Round 2
Reviewer 2 Report
Dear Authors,
Dear authors,
Maybe my comment about the vehicle control was not very clear. What I meant to say is that it is not pointed out in the material and methods, or it is still not clear to me, the use of a control group with the solvent (vehicle) used to dissolve the extracts in the antibacterial test. In the antifungal tests, Acetone 30% was used. Would this be the solvent vehicle? In both tests?
Round 3
Reviewer 2 Report
The authors have now included the control groups adequately and sufficiently for the study. Therefore they have answered my last few questions.
Author Response
Please see the attachment
This manuscript is a resubmission of an earlier submission. The following is a list of the peer review reports and author responses from that submission.
Round 1
Reviewer 1 Report
The manuscript is about the effect of climatic difference on the antibacterial and secondary metabolites of Helichrysum aureonitens Sch. Bip.. The research has been well designed and well conducted. Also, the result would be informative to the readers in such research area.
After author mention the name of bacteria at first please give an abbreviation in parenthesis.
Reviewer 2 Report
The manuscript covers an essential topic in natural remedy research, the seasonal variation of the chemical profiles, and the bioeffective ingredients of the extracts from Helichrysum aureonitens, an important traditional medicinal herb in South Africa.
The study is well-conducted with conclusive data and accurate data interpretation. Furthermore, the authors have stated and concluded well the extent of what they have learned, and they propose proper timing when and under what specific climates to collect the metabolites of their interests. Therefore, I only have minor concerns and suggest a straightforward manuscript publication.
Comments to the Authors
Regarding the Results section, I recommend specifying the titles of the following subsections:
2.1 title: Antibacterial activity of the Helichrysum extracts
2.2 title: Antifungal activity of the Helichrysum extracts
2.3 title: Rainfall and Temperature data of the Farming Areas
Reviewer 3 Report
I reviewed the article entitled "Effect of different climatic regions and seasonal variations on antibacterial and antifungal activity, and chemical profile of Helichrysum aureonitens Sch. Bip."
The article needs a series of minor revisions in the text such as italicization, punctuation, and English proofreading. Apart from this, three important shortcomings came to my attention:
1. No representative voucher specimens from each collection site were deposited in herbaria.
2. There is no evidence that the biological experiments were performed with a sufficient number of replicates and a control group is missing.
3. Finally, the major deficiency, in my opinion, is that the extracts submitted for biological testing (acetone - Antifungal experiment), (methanol/water (70:30) - Antibacterial experiment) are of different chemical composition than the extract prepared for metabolomic analysis, 0.75 mL deuterated methanol (CD3OD) and 0.75 mL, deuterium water (D2O).
Minor points:
1. No quality control method to evaluate extraction (internal standard) and NMR analysis (quality control samples, randomization).
2. Why were the plants dried for 4 months? Is this correct? Is there any justification or recommendation (References) for such a long drying period? Please make this explicit in the text.
3. The methods used to multivariate analysis must be presented in more details. For exemple: which scaling was applied, Pareto or unit variance?
4. Revision of the name of the extracts. For exemple: In several parts of the text it says "methanolic extract" but hydroalcoholic extract was prepared.